

# Fluoroquinolone resistance determinants in carbapenem-resistant *Escherichia coli* isolated from urine clinical samples in Thailand

Parichart Boueroy[1], Peechanika Chopjitt[1], Rujirat Hatrongjit[2], Masatomo Morita[3], Yo Sugawara[4,5], Yukihiro Akeda[3,4], Tetsuya Iida[6], Shigeyuki Hamada[4] and Anusak Kerdsin[1]

[1] Faculty of Public Health, Kasetsart University, Chalermphrakiat Sakon Nakhon Province Campus, Sakon Nakhon, Thailand
[2] Faculty of Science and Engineering, Kasetsart University, Chalermphrakiat Sakon Nakhon Province Campus, Sakon Nakhon, Thailand
[3] Department of Bacteriology I, National Institute of Infectious Diseases, Tokyo, Japan
[4] Japan-Thailand Research Collaboration Center for Infectious Diseases, Research Institute for Microbial Diseases, Osaka University, Osaka, Japan
[5] Antimicrobial Resistance Research Center, National Institute of Infectious Diseases, Tokyo, Japan
[6] Department of Infection Metagenomics, Research Institute for Microbial Diseases, Osaka University, Suita, Osaka, Japan

Corresponding author
Parichart Boueroy,
parichart.bou@ku.th

## ABSTRACT

**Background.** *Escherichia coli* is the most common cause of urinary tract infections and has fluoroquinolone (FQ)-resistant strains, which are a worldwide concern.

**Objectives.** To characterize FQ-resistant determinants among 103 carbapenem-resistant *E. coli* (CREc) urinary isolates using WGS.

**Methods.** Antimicrobial susceptibility, biofilm formation, and short-read sequencing were applied to these isolates. Complete genome sequencing of five CREcs was conducted using short- and long-read platforms.

**Results.** ST410 (50.49%) was the predominant ST, followed by ST405 (12.62%) and ST361 (11.65%). Clermont phylogroup C (54.37%) was the most frequent. The genes *NDM-5* (74.76%) and *CTX-M-15* (71.84%) were the most identified. Most CREcs were resistant to ciprofloxacin (97.09%) and levofloxacin (94.17%), whereas their resistance rate to nitrofurantoin was 33.98%. Frequently, the gene *aac(6′)-Ib* (57.28%) was found and the coexistence of *aac(6′)-Ib* and $bla_{\text{CTX-M-15}}$ was the most widely predominant. All isolates carried the *gyrA* mutants of S83L and D87N. In 12.62% of the isolates, the coexistence was detected of *gyrA*, *gyrB*, *parC*, and *parE* mutations. Furthermore, the five urinary CREc-complete genomes revealed that $bla_{\text{NDM-5}}$ or $bla_{\text{NDM-3}}$ were located on two plasmid Inc types, comprising IncFI (60%, 3/5) and IncFI/IncQ (40%, 2/5). In addition, both plasmid types carried other resistance genes, such as $bla_{\text{OXA-1}}$, $bla_{\text{CTX-M-15}}$, $bla_{\text{TEM-1B}}$, and *aac(6′)-Ib*. Notably, the IncFI plasmid in one isolate carried three copies of the $bla_{\text{NDM-5}}$ gene.

**Conclusions.** This study showed FQ-resistant determinants in urinary CREc isolates that could be a warning sign to adopt efficient strategies or new control policies to prevent further spread and to help in monitoring this microorganism.

## INTRODUCTION

Urinary tract infections (UTIs) are one of the most common bacterial infections with ∼150 million cases per year globally, which pose a large burden on the healthcare system as the associated costs have reached USD 6 billion (*Badamchi et al., 2019*; *Foxman, 2014*). A UTI is one of the most common extraintestinal infections, with 80% of UTIs caused by *Escherichia coli*, which is the target of empirical therapy (*Córdoba et al., 2017*). Fluoroquinolones (FQs) are the drugs of choice to treat UTIs, especially complicated cases and catheter-associated UTIs, since this antibiotic class has a wide spectrum of antimicrobial activities with excellent bioavailability, good oral absorption, and good tissue penetrability (*Sharma, Jain & Jain, 2009*). Although serious side effects associated with fluoroquinolone treatment have been mentioned by the US FDA (*US Food and Drug Administration, 2016*), this antibiotic remains the drug of choice.

The progressive emergence of resistance to fluoroquinolones and other antibiotics commonly used for UTI treatment has been observed in several countries over the last few years (*Córdoba et al., 2017*; *Lee, Lee & Choe, 2018*; *Cunha et al., 2016*). The resistance rate against FQ has recently increased, particularly among the Enterobacterales, especially *E. coli* (*Faine et al., 2022*; *Kibwana et al., 2023*). FQ resistance in *E. coli* showed that the mutations in the quinolone resistance-determining regions (QRDRs) of the chromosomal genes *gyrA*, *parC*, and *parE* lead to alteration of the target proteins (DNA gyrase and topoisomerase IV) of these antimicrobial agents and the operons of the endogenous transmembrane efflux pump AcrAB-TolC (*marR* and *acrR*) (*Chen, Erickson & Meng, 2020*). The efflux pump (*oqxAB*), a variant of the aminoglycoside-modifying enzyme *aac(6′)-Ib-cr* and qnr determinants (*qnrA*) that encode DNA gyrase protection proteins, was also identified to have the potential to reduce the susceptibility to FQs and lead to resistance in *E. coli* (*Cheng et al., 2020*).

The most common mechanisms are the mutation of the gene that encodes type II topoisomerases (DNA gyrase and topoisomerase IV), which are enzymes that are essential for DNA replication and alter the fluoroquinolone binding affinity of the enzyme. It inhibits the activity of DNA gyrase and topoisomerase IV, which are enzymes that are essential for DNA replication (*Hopkins, Davies & Threlfall, 2005*).

In Thailand, there is insufficient data about FQ-resistant *E. coli*. In 2015, we began conducting the Emerging Antimicrobial Resistant Bacterial Surveillance Program (EARB). In this program, we focused on carbapenem-resistant Enterobacterles (CRE) and colistin-resistant Enterobacterles isolated from patients in 11 hospital networks in Thailand (*Takeuchi et al., 2022*; *Boueroy et al., 2022*).

Our previous CRE study showed *K. pneumoniae* (577 isolates) and *E. coli* (170 isolates) accounted for 97.5% of all CRE isolates (n = 766) (*Takeuchi et al., 2022*). In *Takeuchi et al. (2022)*, *E. coli* isolates were most frequently isolated from urine (58.24%, 99/170),

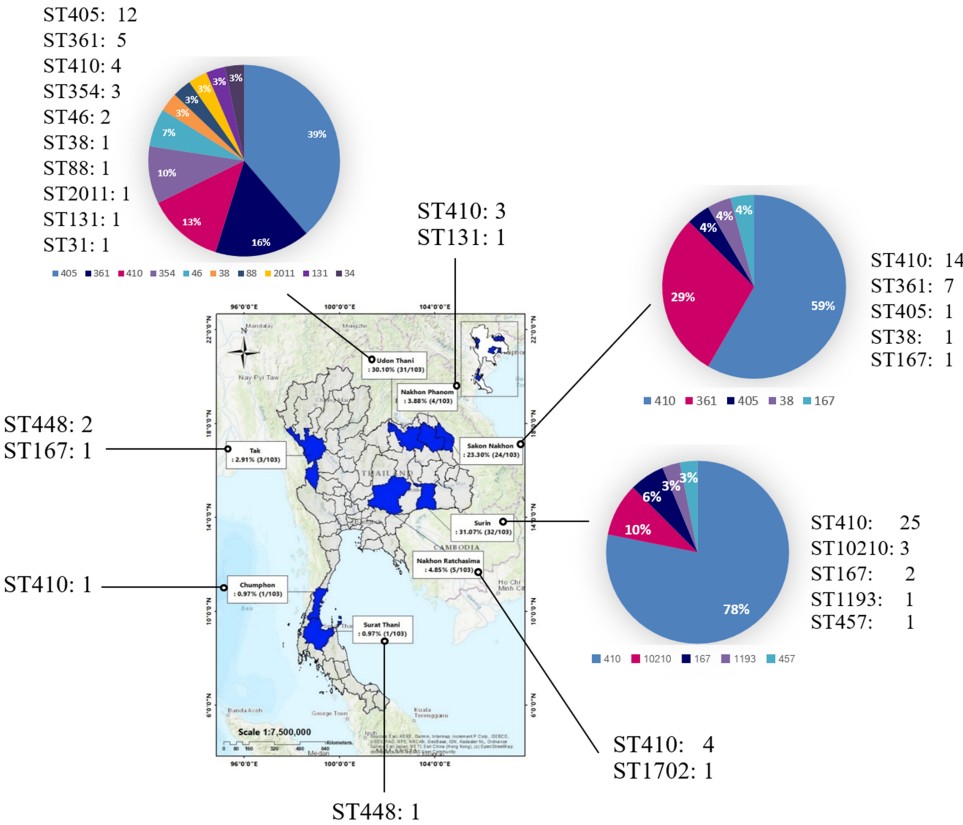

**Figure 1  Provinces where samples were collected between 2015 and 2020.** Samples were collected from eight provinces nationwide: Surin, Udon Thani, Sakon Nakhon, Nakhon Ratchasima, Nakhon Phanom, Tak, Surat Thani, and Chumphon. (A geographical information system (GIS) software QGIS (version 2.18.28) was used to create a study map). The total number of isolates is shown in the map, and the number and percentage of ST is presented in the pie chart.

followed by blood (13.53%, 23/170). *Takeuchi et al. (2022)* mentioned the prevalence of carbapenemase genotypes and replicon types of carbapenemase gene-harboring plasmids but did not analyze FQ and other antimicrobial resistance determinants. The present study aimed to determine the antimicrobial resistance determinants (other than carbapenem), including fluoroquinolone resistance, extended-spectrum beta-lactamase production, virulence, and genotypes, among CREcs obtained from the urine of patients in Thailand during 2015–2020.

# MATERIALS & METHODS

## Bacterial isolates

The present study used 103 urine isolates of CREc during 2015–2020, comprising 88 isolates from a previous study (*Takeuchi et al., 2022*) and 15 isolates from the present study from eight provinces in Thailand under the EARB program (Fig. 1 and Table S1).

CREc definition in the present study revealed *E. coli* strains that are resistant to at least one of the carbapenem antibiotics (ertapenem, meropenem, doripenem, or imipenem) or produce a carbapenemase, an enzyme that can destroy carbapenem antibiotics.

## Ethical approval

Ethical approval was obtained from the Ethics Committee of Osaka University Graduate School of Medicine, Osaka, Japan, with approval number 14468-5. The present study was conducted following the principles of the Declaration of Helsinki; the need for informed consent was waived.

## Antimicrobial susceptibility testing

All isolates were subjected to antimicrobial susceptibility testing using the broth microdilution method according to 2022 Clinical and Laboratory Standards Institute guidelines (*Clinical Laboratory Standard Institute, 2022*). The broth microdilution method was conducted using cation-adjusted Mueller–Hinton broth (Becton, Dickinson and Company; Sparks, MD, USA) for antimicrobial susceptibility testing of ciprofloxacin, levofloxacin, and carbapenem. Susceptibility to nitrofurantoin (NFT) was carried out *via* a disk diffusion technique and was interpreted based on *Clinical Laboratory standard institute (2022)*. *E. coli* ATCC 25922 was used for quality control.

## Detection of ESBL production

The production of ESBL was tested for the 103 urinary CREc isolates using the combined disk method with cefotaxime (30 μg) and ceftazidime (30 μg) with or without clavulanic acid (10 μg) (*Clinical Laboratory Standard Institute, 2022*). An increase in the zone size ≥5 mm for cefotaxime and ceftazidime with or without clavulanic acid was considered to indicate ESBL production (*Clinical Laboratory Standard Institute, 2022*). *E. coli* ATCC 25922 was used as a negative control.

## Biofilm formation assay

The biofilm production of the urinary CREc isolates was determined using the Congo red agar (CRA) method, as previously described (*Tajbakhsh et al., 2016*). Briefly, the CREc isolates were cultured on CRA made of brain heart infusion agar with 36 g/L sucrose and Congo red dye 0.8 g/L, and they were incubated at 37 °C for 24–48 h. The biofilm production was characterized based on six color tones of colonies: very black, black, and almost black (interpreted as strong, moderate, and weak biofilm producers, respectively) and bordeaux, red, and very red (reported as non-biofilm producers).

## Whole-genome sequencing and genome assembly

Bacterial DNA was extracted using the Applied Biosystems™ MagMAX™ DNA Multi-Sample Ultra 2.0 Kit (Thermo Fisher Scientific, Waltham, MA, USA) according to the manufacturer's instructions. All 103 isolates were sequenced using the short-read sequencing Illumina platform, HiSeq 3000 (Illumina, San Diego, CA, USA) (*Takeuchi et al., 2022*).

Complementarily, five isolates (nos. AMR0278, C032, C300, C359, and C439) were selected for long-read sequencing using the Oxford Nanopore Technologies (ONT)

platform (Oxford Nanopore Technologies, Oxford, UK) based on their representative strains for each main branch of the dendrogram constructed using Illumina assembled data (Fig. S1). Library preparation and sequencing were determined as described in *Boueroy et al. (2022)*. Raw data were demultiplexed using Guppy v3.4.5 (ONT), specifying the high-accuracy model (–c dna_r9.4.1_450bps_hac.cfg). The ONT adapters were trimmed using Porechop v0.2.4 (https://github.com/rrwick/Porechop).

To obtain complete genomes, hybrid assemblies using the Illumina and ONT data were generated using Unicycler v0.4.8 (*Wick et al., 2017*; *Chen, Erickson & Meng, 2020*), and the quality of the assembly checked using QUAST v5.0.2. (*Gurevich et al., 2013*). Genome sequences were annotated using Prokka v1.14.6 (*Seemann, 2014*). The quality assemblies of the five isolates were the mean total sequence length (bp), N50, and the GCcontent (%), being 5103581.6, 4865394.2, and 50.62, respectively (Table S2).

## Bioinformatics analysis

The multilocus sequence typing (MLST), serotype, and *fimH* type were analyzed using the Center for Genomic Epidemiology website (http://www.genomicepidemiology.org/). The presence of antimicrobial resistance genes and virulence genes were determined using the ResFinder v4.0 (*Bortolaia et al., 2020*), CARD (*Alcock et al., 2023*), and ecoli_vf databases. ABRicate v0.3 (https://github.com/tseemann/abricate) was used to scan assemblies for genes related to antimicrobial resistance and virulence and plasmid replicons compared to the ResFinder, ecoli_VF (https://github.com/phac-nml/ecoli_vf), and PlasmidFinder v2.1 (*Carattoli et al., 2014*) databases. Phylogenetic group identification was carried out using ClermonTyping 21.03 (http://clermontyping.iame-research.center/index.php) (*Clermont et al., 2013*). PubMed was searched for primary research articles describing mobile genetic elements (MGEs) carrying the same AMR genes in plasmid to identify putative homologous MGEs.

The QRDR mutations in *gyrA*, *gyrB*, *parC*, and *parE* were determined by performing a multiple sequence alignment of the protein sequence *via* Clustal Omega (https://www.ebi.ac.uk/Tools/msa/clustalo/) using the genes of *E. coli* strain K-12 (MG1655) as a reference.

We aligned the protein sequences of the IncFI plasmid carrying *NDM-5* with homologous plasmids reported from China (MF156715.1), France (LR595692.1), and Myanmar (AP019191.1). Additionally, we aligned the protein sequences of the ColKP3-type plasmid carrying *bla*OXA-181 and *qnrS* (no. AMR0278) with homologous plasmids reported from the Netherlands (NZ_CP068910.1, NZ_CP068959.1, NZ_CP068881.1, NZ_CP068958.1, and NZ_CP068938.1), Switzerland (NZ_CP048332.1, NZ_CP048327.1, NZ_CP048321.1, and NZ_CP048325.1), Ghana (NZ_CP081309.1), Egypt (NZ_CP048918.1), France (NZ_LR595693.1), China (NZ_CP043335.1), and the United States (NZ_CP034284.1). The annotated MGEs were aligned using clinker and clustermap.js v.0.021 (*Gilchrist & Chooi, 2021*). The plasmid maps of strain no. AMR0278 was generated using Proksee (*Grant, Arantes & Stothard, 2012*).

## Phylogenetic analysis

Panaroo v1.2.10 (https://github.com/gtonkinhill/panaroo) (*Tonkin-Hill et al., 2020*) was used to reconstruct the pangenome by grouping genes from the annotated assemblies into homology groups. The core-genome alignment was applied to build the phylogeny with IQ-TREE multicore version 2.2.0.3 (*Nguyen et al., 2015*). Trees were visualized using the interactive tree of life (iTOL) v6.5 (*Letunic & Bork, 2021*).

## Pathotyping of CREc

Extraintestinal pathogenic *E. coli* (ExPEC) strains were identified following previously described criteria; they were classified as positive if ≥2 of the following five genes: *papA*, and/or *papC*, *sfa/focDE*, *afa/draBC*, *iutA*, and *kpsMTII* (*Johnson et al., 2003*). Uropathogenic *E. coli* (UPEC) was determined following previously reported criteria; specifically, if ≥3 of the four key marker genes (*fyuA*, *chuA*, *vat*, and *yfcV*) were present (*Spurbeck et al., 2012*). Avian pathogenic *E. coli* (APEC) were classified based on the presence or absence of all of the 13 virulence-associated genes: *fliCH4*, *arpA*, *aec4*, *ETT22*, *frzorf4*, *fyuA*, *iha*, *ireA*, *iroN*, *iutA1*, *papA*, *tsh*, and *vat* (*Lucas et al., 2022*).

## Statistical analysis

Data were analyzed using SPSS version 26.0 (Chicago, IL, USA). Fisher's exact test was used to establish the association between biofilm formation ability and adhesion factor genes in the 103 urine isolates of CREc. $P < 0.05$ was considered statistically significant.

# RESULTS AND DISCUSSION

## Genotypic profiles of CREc urinary isolates (FQ-CREc)

Overall, 50.49% (52/103) of the isolates were ST410, followed by ST405 (12.62%, 13/103) and ST361 (11.65%, 12/103) as shown in Table S1. The 52 CREc ST140 isolates from the urine clinical samples were mainly isolated from Surin (25 isolates), Sakon Nakhon (14 isolates), Nakhon Ratchasima (four isolates), and Udon Thani (four isolates) provinces, Thailand (Fig. 1).

The CREc isolates were classified into phylogroups C (54.37%, 56/103), A (19.42%, 20/103), D (14.56%, 15/103), F (4.85%, 5/103), B1 (3.88%, 4/103), and B2 (2.91%, 3/103), respectively (Table S1). As shown in Table 1, the Clermont phylogroup C contained mostly ST410, ST12010, and ST88, whereas phylogroup A contained STs 34, 46, 167, 361, and 1702, while phylogroup D consisted of STs 38 and 405. Conversely, *FimH* typing demonstrated that the majority of these CREc isolates were *fimH* 24 (53.40%, 55/103), whereas O8:H9 or H9 (53.40%, 55/103) were the major serotypes in the present study.

To determine genetic relationships, a core-genome SNP-based phylogeny of the 103 CREc genomes is shown in Fig. 2. Most CREc urinary isolates were divided into six large clusters based on the main branches of the tree. The first cluster comprised ST361. The second cluster contained ST46. The third cluster consisted of ST34, ST167, and ST1702. The fourth cluster contained ST1193, ST131, ST2011, ST457, and ST354. The fifth cluster consisted of ST38 and ST405. The last cluster contained mainly ST410 strains and ST88 and ST448 (Fig. 2).

**Table 1  Antimicrobial resistance determinants of FQ-CREc urine isolates in Thailand.**

| MLST | Clermont phylogroup | $bla_{VEB-type}$ | qnr | aac(6)-Ib-cr | gyrA | gyrB | parC | parE | CPFX | LVFX | Biofilm | Profiles | N |
|---|---|---|---|---|---|---|---|---|---|---|---|---|---|
| 410 | C | – | – | – | + | – | + | + | R | R | Strong biofilm | 1 | 9 |
| 410 | C | – | – | + | + | – | + | + | R | R | non biofilm producers | 2 | 7 |
| 410 | C | – | – | + | + | – | + | + | R | R | non biofilm producers | 3 | 6 |
| 410 | C | – | – | + | + | – | + | + | R | R | weak biofilm | 4 | 3 |
| 410 | C | – | – | + | + | – | – | + | R | R | non biofilm producers | 5 | 2 |
| 410 | C | – | – | + | + | – | – | + | R | R | non biofilm producers | 6 | 2 |
| 410 | C | – | – | + | + | – | + | – | R | R | non biofilm producers | 7 | 1 |
| 410 | C | – | qnrS 1 | + | + | – | + | + | R | R | non biofilm producers | 8 | 1 |
| 410 | C | – | – | + | + | – | + | + | R | R | Strong biofilm | 9 | 1 |
| 410 | C | – | – | – | + | – | + | + | R | R | Strong biofilm | 10 | 1 |
| 410 | C | – | – | – | + | – | + | + | R | R | non biofilm producers | 11 | 1 |
| 410 | C | – | - | – | + | – | + | + | R | R | weak biofilm | 12 | 1 |
| 410 | C | – | – | + | + | – | + | + | R | R | weak biofilm | 13 | 1 |
| 410 | C | – | - | + | + | – | + | + | R | R | Strong biofilm | 14 | 1 |
| 410 | C | – | – | – | + | – | + | + | R | R | weak biofilm | 15 | 1 |
| 410 | C | – | – | + | + | – | + | + | R | R | weak biofilm | 16 | 1 |
| 410 | C | – | – | + | + | – | + | + | R | R | moderate biofilm | 17 | 1 |
| 410 | C | – | – | + | + | – | + | + | R | R | weak biofilm | 18 | 1 |
| 410 | C | – | – | + | + | – | + | + | R | R | non biofilm producers | 19 | 1 |
| 410 | C | – | qnrS 1 | – | + | – | + | + | R | R | non biofilm producers | 20 | 1 |
| 410 | C | – | – | + | + | – | + | – | R | R | non biofilm producers | 21 | 1 |
| 410 | C | – | – | + | + | – | + | + | R | R | Strong biofilm | 22 | 1 |
| 410 | C | VEB-1 | - | + | + | – | + | + | R | R | non biofilm producers | 23 | 1 |
| 410 | C | – | – | + | + | – | + | + | R | R | Strong biofilm | 24 | 1 |
| 410 | C | – | - | + | + | – | + | + | R | R | Strong biofilm | 25 | 1 |
| 410 | C | – | - | + | + | – | + | + | R | R | non biofilm producers | 26 | 1 |
| 410 | C | VEB-1 | qnrA1 | + | + | – | + | + | R | R | non biofilm producers | 27 | 1 |
| 410 | C | – | qnrB17,qnrS1 | + | + | – | + | + | R | R | moderate biofilm | 28 | 1 |
| 410 | C | – | qnrS1 | + | + | – | + | + | R | R | non biofilm producers | 29 | 1 |
| 405 | D | – | – | + | + | + | + | + | R | R | weak biofilm | 1 | 2 |
| 405 | D | – | – | + | + | + | + | + | R | R | non biofilm producers | 2 | 2 |
| 405 | D | – | – | – | + | + | + | + | R | R | weak biofilm | 3 | 2 |
| 405 | D | – | – | – | + | + | + | + | R | R | non biofilm producers | 3 | 1 |
| 405 | D | – | – | + | + | + | + | + | R | R | Strong biofilm | 4 | 1 |
| 405 | D | – | – | + | + | + | – | + | R | R | weak biofilm | 5 | 1 |
| 405 | D | – | – | + | + | + | + | + | R | R | weak biofilm | 6 | 1 |
| 405 | D | – | – | – | + | + | – | + | R | R | non biofilm producers | 7 | 1 |
| 405 | D | – | – | + | + | + | + | + | R | R | weak biofilm | 8 | 1 |
| 405 | D | – | – | – | + | + | – | + | R | R | weak biofilm | 9 | 1 |
| 361 | A | – | – | – | + | – | + | – | R | R | non biofilm producers | 1 | 5 |
| 361 | A | – | – | – | + | – | + | – | R | R | non biofilm producers | 2 | 3 |

**Table 1** (*continued*)

| MLST | Clermont phylogroup | $bla_{VEB-type}$ | qnr | aac(6)-Ib-cr | gyrA | gyrB | parC | parE | CPFX | LVFX | Biofilm | Profiles | N |
|---|---|---|---|---|---|---|---|---|---|---|---|---|---|
| 361 | A | – | – | + | + | – | + | – | R | R | non biofilm producers | 3 | 1 |
| 361 | A | – | – | – | + | + | + | – | R | R | non biofilm producers | 4 | 1 |
| 361 | A | – | – | + | + | – | + | – | R | R | non biofilm producers | 5 | 1 |
| 361 | A | – | – | – | + | – | – | – | R | R | non biofilm producers | 6 | 1 |
| 167 | A | – | – | – | + | – | + | + | R | R | weak biofilm | 1 | 1 |
| 167 | A | – | qnrS1 | – | + | – | + | + | R | R | Strong biofilm | 2 | 1 |
| 167 | A | – | – | – | + | – | + | + | R | R | weak biofilm | 3 | 1 |
| 167 | A | – | - | + | + | – | + | + | R | R | weak biofilm | 4 | 1 |
| 448 | B1 | – | – | + | + | – | + | + | R | R | weak biofilm | 1 | 1 |
| 448 | B1 | – | – | + | + | – | + | + | R | R | non biofilm producers | 2 | 1 |
| 448 | B1 | – | – | – | + | – | + | + | R | R | weak biofilm | 3 | 1 |
| 448 | B1 | – | – | – | + | – | + | + | R | R | non biofilm producers | 4 | 1 |
| 354 | F | – | – | – | + | + | + | + | R | R | Strong biofilm | 1 | 1 |
| 354 | F | – | – | – | + | + | + | + | R | R | moderate biofilm | 2 | 1 |
| 354 | F | – | qnrB6 | – | + | + | + | + | R | R | moderate biofilm | 3 | 1 |
| 10210 | C | – | – | + | + | – | + | + | R | R | non biofilm producers | 1 | 1 |
| 10210 | C | – | – | + | + | – | + | + | R | R | Strong biofilm | 2 | 1 |
| 10210 | C | – | – | + | + | – | – | – | R | R | non biofilm producers | 3 | 1 |
| 46 | A | – | – | + | + | – | + | + | R | R | non biofilm producers | 1 | 1 |
| 46 | A | – | – | + | + | – | + | + | R | R | weak biofilm | 2 | 1 |
| 38 | D | – | – | + | + | – | + | – | S | S | moderate biofilm | 1 | 1 |
| 38 | D | – | – | – | + | – | + | + | R | R | weak biofilm | 2 | 1 |
| 131 | B2 | – | – | + | + | – | + | + | R | R | moderate biofilm | 1 | 1 |
| 131 | B2 | – | – | – | + | – | + | + | R | R | Strong biofilm | 2 | 1 |
| 2011 | F | – | – | + | + | + | + | + | R | R | moderate biofilm | 1 | 1 |
| 457 | F | – | – | – | + | + | + | + | R | R | moderate biofilm | 1 | 1 |
| 1193 | B2 | – | – | – | + | – | + | + | R | R | moderate biofilm | 1 | 1 |
| 1702 | A | – | – | – | + | – | + | + | R | R | non biofilm producers | 1 | 1 |
| 34 | A | – | – | + | + | – | – | – | S | S | non biofilm producers | 1 | 1 |
| 88 | C | – | – | – | + | + | | + | | S | S | non biofilm producers | 1 | 1 |

**Notes.**

Abbreviation: MLST, multilocus sequence typing; CPFX, ciprofloxacin; LVFX, levofloxacin; +, positive; -, negative; R, resistance; S, sensitivity.

The present study revealed that the CREc urinary isolates mainly belonged to the phylogroup C, which are considered to be associated with commensal status or intestinal pathotypes (*Tenaillon et al., 2010*). Nevertheless, the UTI *E. coli* isolates were significantly more frequent in the phylogroups B2 and D (*Amarsy et al., 2019*). Our study showed that phylogroup D had second prevalence and phylogroup B2 had rare status. The Clermont phylogroup C contained mainly ST410. This ST is an emerging and international high-risk clone worldwide, associated with a large number of clinical infections (*Roer et al., 2018*). In Thailand, ST410 has been identified in clinical isolates that co-harbored *mcr* and $bla_{NDM}$ (*Boueroy et al., 2022*; *Paveenkittiporn et al., 2021*).
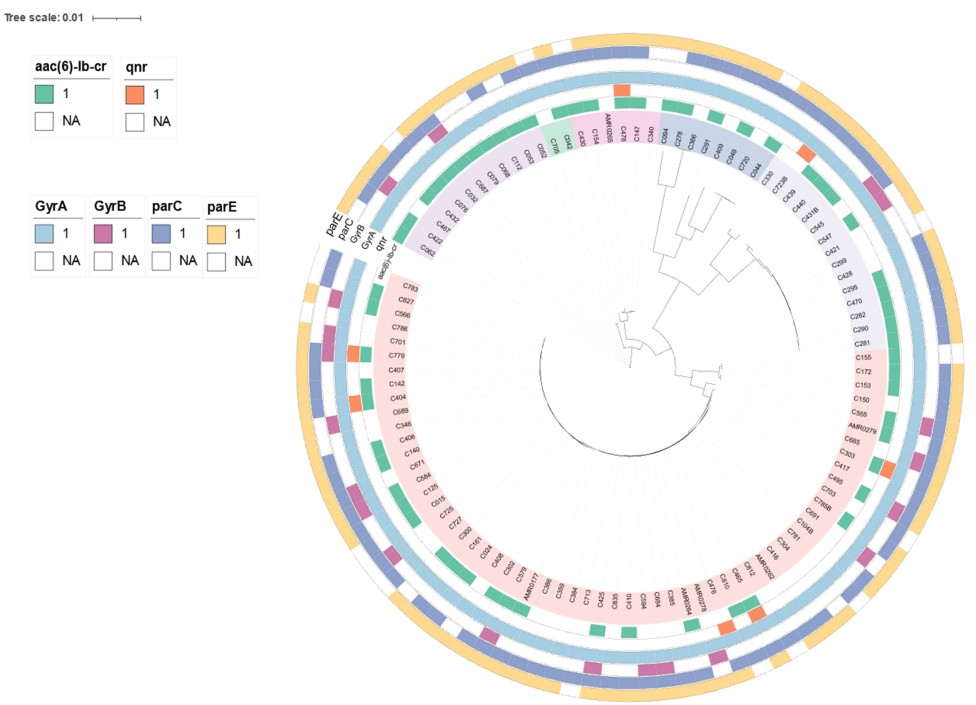

**Figure 2** **The existence of PMQR genes (*aac(6′)-Ib* and *qnr*) and QRDR mutations (*gyrA, gyrB, parC,* and *parE*) in CREc urinary isolates.** Core-genome maximum-likelihood phylogeny based on 103 CREc isolates genomes reconstructed using Panaroo and IQ-TREE. The tree was visualized using iToL.

## Pathotyping, virulence genes, and biofilm formation of urinary CREc

Pathotyping identified only 17.48% (18/103) ExPEC of urinary CREc isolates. One (0.97%, 1/103) CREc isolate was identified as UPEC (≥3 UPEC virulence factors) (Table 2). No APEC was found in the present study. Although most urinary CREc in this study could not be classified, they seem to be commensal or non-pathogenic pathotypes. Among the three investigated virulence factor classes, 20.39% (21/103) of the isolates were identified as having two UPEC-associated markers (Table 2). All 103 urinary CREc isolates carried one or more APEC-associated markers: *fimC* (91.26%, 94/103), *fyuA* (57.28%, 59/103), *irp2* (47.57%, 49/103), *iss* (24.27%, 25/103), *iucD* (17.48%, 18/103), *papC* (10.68%, 11/103), *astA* (2.91%, 3/103), and *vat* (0.97%, 1/103) (Table 2).

In total, 37 virulence genes were examined and categorized as adhesion molecules, invasion genes, toxins, iron uptake molecules, autotransporter systems, and protection factors, according to the genotypic profiles of the 103 urinary CREc isolates (Table S3). Overall, *fim* (100%), *csg* (100%), and *ibeABC* (100%) were the virulence genes with the highest distributions among isolates, followed by *ecp* (98.06%), *espl* (97.09%), *hlyE* (97.09%), *eae* (78.64%), *cah* (67.96%), *upaG/ehaG* (65.05%), *tia* (49.51%), and *fyuA* (47.57%) (Table S3). This was not surprising because these virulence factors, such as type 1 fimbriae (*fim*) and the major pilus subunit (*ecpA*), have also been reported in commensal *E. coli* (*Pusz et al., 2014*; *Blackburn et al., 2009*).

**Table 2 Genotypic and phenotypic virulence factors of urinary CREc.**

| Virulence factor classes | Virulence factors | Prevalence (%) of isolates | No. of positive/ No. of total |
|---|---|---|---|
| Extraintestinal pathogenic *E. coli* (ExPEC) | *papAH* | 15.53 | 16/103 |
| | *papC* | 10.68 | 11/103 |
| | *sfa/focDE* | 0 | 0/103 |
| | *afa/draBC* | 3.88 | 4/103 |
| | *kpsM* | 18.45 | 19/103 |
| | *iutA* | 8.74 | 9/103 |
| | ≥2 ExPEC VFs | 17.48 | 18/103 |
| | *vat* | 0.96 | 1/103 |
| Uropathogenic *E. coli* (UPEC) | *fyuA* | 57.69 | 59/103 |
| | *chuA* | 22.12 | 22/103 |
| | *yfcV* | 0 | 0/103 |
| | ≥3 UPEC VFs | 0.97 | 1/103 |
| | *fliCH4* | 0 | 0/103 |
| | *arpA* | 0 | 0/103 |
| | *aec4* | 0 | 0/103 |
| | *ETT22* | 0 | 0/103 |
| | *frzorf4* | 0 | 0/103 |
| Avian pathogenic *E. coli* (APEC) | *fyuA* | 57.28 | 59/103 |
| | *iha* | 0 | 0/103 |
| | *ireA* | 0 | 0/103 |
| | *iroN* | 0.97 | 1/103 |
| | *iutA1* | 8.74 | 9/103 |
| | *papA* | 15.53 | 16/103 |
| | *tsh* | 0 | 0/103 |
| | *vat* | 0.97 | 1/103 |
| | 13 APEC | 0 | 0 |

**Notes.**
Abbreviation: VFs, virulence factors.

Almost all the ExPEC isolates in the presents study belonged mainly to the phylogenetic groups B2 and D and carried various virulence factors compared to the other phylogenetic groups (Table S4). Concordance with another report identified a higher number of virulence genes in phylogroups B2 and D compared to other groups in UTI *E. coli* isolates (*Lee & Moon, 2016*). In contrast, most urinary CREc isolates in the present study were not classified as any pathotype, which was consistent with them being Clermont types C and A (73.79%; 76/103).

In the biofilm assay, 50.49% (52/103) of the urinary CREc isolates produced weak-to-strong biofilm formation (Tables 1 and S5). Among these isolates, 20%, 9%, and 23% were strong, moderate, and weak biofilm producers, respectively. More than one-half (71.84%; 74/103) of these isolates could also be classified as weak and non-biofilm formation. In agreement with our findings, biofilm formation has been reported as low in prevalence in commensal *E. coli* strains (*Karam, Habibi & Bouzari, 2018*). According to another report,

biofilm formation has greater carriage of adhesin genes (*Fim*, *Pap*, *Sfa*, and *Afa*) compared to non-biofilm formers (*Katongole et al., 2020*). Nevertheless, we did not observe any relationship between biofilm formation ability and the carriage of any adhesin genes (Table S6). Consistent with another report, the biofilm formation in carbapenem-resistant *E. coli* from the UTIs did not reveal any significant difference in the carriage rate of fimbriae genes (*Chen et al., 2023*).

## Antimicrobial susceptibility and resistant genes in urinary CREc isolates

Among the 103 urinary CREc isolates, resistance levels of ciprofloxacin and levofloxacin were 97.09% (100/103) and 94.17% (97/103), respectively. Almost all isolates resisted FQs, with MICs, ranging from 2 to >32 μg/ml (Tables 2 and S5). The ESBL screening test showed that 77.67% of the isolates were ESBL producers, among which, 97.50% were also resistant to FQs (Tables 1 and S3). The resistance rate of NFT was 33.98% (35/103) in the CREc urinary isolates (Tables 1 and S3).

Carbapenems are the drug of choice for complicated UTIs caused by multidrug-resistant Enterobacteriaceae, especially those due to ESBL-producing *E. coli* (*Amladi et al., 2019*). Although FQs are the drug of choice to treat UTIs treatment, FQ resistance is an increasing issue that causes concern in several countries, including Greece, Senegal, and Saudi Arabia (*Yang et al., 2010*; *Chaniotaki et al., 2004*). National Antimicrobial Resistance Surveillance Thailand (http://narst.dmsc.moph.go.th/) reported that *E. coli* urinary isolates had a high percentage of resistance against ciprofloxacin (67.6%) and levofloxacin (66.6%).

Our urinary CREc isolates exhibited high resistance to ciprofloxacin (97.09%) and levofloxacin (94.17%). The high frequency among FQ-resistant determinants in urinary CREc isolates in our study raises serious concerns and the need to identify alternative antibiotics for UTI therapy. One choice is nitrofurantoin as a suitable candidate for the treatment of UTIs caused by multidrug-resistant pathogens (*Munoz-Davila, 2014*). Several studies have reported a low resistance rate of urinary *E. coli* to NFT, including 18.4% in Turkey (*Pullukçu et al., 2007*), 6.2% in Spain, and 3.7% in England (*Farrell et al., 2003*; *Gobernado et al., 2007*). Resistance rates to NFT in the CREc urinary isolates in the present study showed a low prevalence (33.98%) compared to other antibiotics. The resistance rate of NFT remained virtually unchanged, suggesting that it may be an important and economical option for UTI treatment (*McKinnell et al., 2011*).

Among the 103 isolates, 82 carried $bla_{CTX-M-type}$ ESBL genes (79.61%), with $bla_{CTX-M-15}$ being predominant in urinary CREc isolates (74/82 (90.24%) (Table S7). In addition, $bla_{CTX-M-27}$ was detected in ST131 (no. C366) and ST38 isolates (nos. C330 and C723B). The coexistence of $bla_{CTX-M-55}$ and $bla_{CTX-M-24}$ was identified in three CREs isolates with ST354. The carbapenemase genes identified were $bla_{NDM-5}$ (74.76%, 77/103), followed by $bla_{NDM-1}$ (17.48%, 18/103) (17.48%, 18/103) (Table S7). Other β-lactamase genes were detected in these urinary CREc isolates, such as $bla_{TEM-1B}$ (84.47%, 87/103), $bla_{CMY}$ (67%, 69/103), $bla_{VEB}$ (1.94%, 2/103), and $bla_{TEM-150}$ (0.975, 1/103) (Table S7).

In this study, CREc urinary isolates harboring *blaNDM-5* and *blaCTX-M-15* were the majority of carbapenemase and ESBL genes, respectively. Other studies have revealed that

NDM, especially $bla_{NDM-5}$, is the main carbapenemase gene in CRE in Thailand (*Takeuchi et al., 2022*; *Paveenkittiporn et al., 2021*). The most widely distributed $bla_{CTX-M}$ enzyme worldwide is $bla_{CTX-M-15}$, which is commonly found in *E. coli* isolates causing UTI (*Price et al., 2013*).

The PMQR genes were broadly identified among the urinary CREc isolates, with *aac(6′)-Ib* being present in 60 isolates (58.25%), *qnrA1* in one isolate (0.97%), *qnrB6* in one isolate (0.97%), *qnrS1* in four isolates (3.88%), and *qnrB17+qnrS1* in one isolate (0.97%) (Table 2 and Fig. 2). The coexistence of the *qnrA1* and *aac(6′)-Ib*, or *qnrS1* and *aac(6′)-Ib*, and *qnrB17* and *qnrS1* genes was detected in the ST410 lineage (Table 2). The coexistence of the $bla_{CTX-M}$ and *PMQR* genes was identified in 57 (55.34%) of the CREc isolates. The coexistence of $bla_{TEM-1B}$ and *aac(6′)-Ib* was the most widely distributed resistance genotype, which was observed in 59 (57.28%) of the isolates (Table 2). The coexistence of $bla_{TEM-150}$ and *qnrB6* was identified in one CRE isolate with ST354.

Analysis of the QRDR mutations in *gyrA*, *gyrB*, *parC*, and *parE* revealed that all isolates (100%) carried the *gyrA*- resistant variants S83L (99.03%, 102/103) and D87N (97.09%, 100/103) (Table 3). Resistance due to *gyrB* variants was identified in 20/103 (19.42%) of the isolates with the A618T variant being the most frequent substitution (75%, 15/20). The *parC* resistance-associated substitution was identified in 92/103 (89.32%) of the isolates with the substitution S80I being widely prevalent (94.57%, 87/92) (Table 3). The resistance variants in *parE* were detected in 86/103 (83.50%) of the isolates with the substitution S458A being the most common (86.05%, 74/86) (Table 3). The coexistence of mutations in the *gyrA*, *gyrB*, *parC*, and *parE* genes was detected in 14.56% (15/103) of the isolates, specifically in the ST405 (66.67%, 10/15), ST354 (20%, 3/15), ST457 (6.67%, 1/15), and ST2011 (6.67%, 1/15) lineages (Tables 1 and 3). Among these 15 isolates, all were resistant to FQ (100%, 13/13), were ESBL producers (86.67%, 13/15), and produced biofilms (80%, 12/15) (Tables 3 and S5).

The occurrence of the QRDR mutations is common in FQ-resistant *E. coli* isolated from urine since the mutations play a significant role in conferring quinolone resistance. Mutations in *gyrA* are the primary cause of FQ resistance in Gram-negative clinical isolates (*Varughese et al., 2018*). In the present study, the QRDR *gyrA* S83L and D87N variants were the most common (97.09%), followed by *parC* S80I (84.47%), *parE* S458A (72.82%), and *gyrB* A618T (14.56%). Similarly, the mutations in *gyrA* (S83L, D87N, or D87Y), *parC* (E84V and S80I) *parE* (I529L, L416F, I444F, S458T, D475E, and S458A) have been identified in FQ-resistant *E. coli* clinical isolates in several other studies (*Lindgren et al., 2005*; *Kim, Kim & Lee, 2022*; *Shigemura et al., 2012*).

In the present study, the *aac(6′)-Ib-cr* gene was predominant (58%, 58/100) among the ciprofloxacin-resistant CREc isolates. Similarly, this gene was highly frequent in ciprofloxacin-resistant *E. coli* from UTI patients in Nigeria (*Eghieye et al., 2020*). In contrast, *qnrA*, *qnrB*, and *qnrS* were reported as highly prevalent in other studies (*Eghieye et al., 2020*; *Ramírez-Castillo et al., 2018*) but were low in the present study. Furthermore, the *oqxAB* gene was identified in 5.92% of the FQ non-susceptible *E. coli* isolated from UTI patients in Taiwan (*Kuo et al., 2022*); however, this was not identified in the present study.

**Table 3** The QRDR mutations of FQ-CREc from urine clinical sample in Thailand.

| ST | GyrA | GyrB | parC | parE | Biofilm formations | N |
|---|---|---|---|---|---|---|
| 410 | S83L,D87N | - | S80I | S458A | non biofilm producers | 15 |
| 410 | S83L,D87N | - | S80I | S458A | Strong biofilm | 13 |
| 410 | S83L,D87N | - | S80I | S458A | weak biofilm | 8 |
| 410 | S83L,D87N | - | S80I, D475R | S458A | non biofilm producers | 6 |
| 410 | S83L,D87N | - | - | S458A | non biofilm producers | 4 |
| 410 | S83L,D87N | - | S80I | - | non biofilm producers | 2 |
| 410 | S83L,D87N | - | S80I, D475R | S458A | Strong biofilm | 2 |
| 410 | S83L,D87N | - | S80I | S458A | moderate biofilm | 2 |
| 405 | S83L,D87N,D678E | A618T, T653A, I663V | S80I, D475R | S458A | weak biofilm | 3 |
| 405 | S83L,D87N,D678E | A618T, T653A, I663V | S80I, D475R | S458A | non biofilm producers | 2 |
| 405 | S83L,D87N,D678E | A618T, T653A, I663V | S80I | S458A | weak biofilm | 2 |
| 405 | S83L,D87N,D678E | I663V | S80I | S458A | non biofilm producers | 1 |
| 405 | S83L,D87N,D678E | A618T, T653A, I663V | A192V, Q481H | V136I, I529L | Strong biofilm | 1 |
| 405 | S83L,D87N,D678E | A618T, T653A, I663V | | S458A | weak biofilm | 1 |
| 405 | S83L,D87N,R237H | A618T, T653A, I663V | | S458A | non biofilm producers | 1 |
| 405 | S83L,D87N,D678E | A618T, T653A, I663V | A192V, Q481H | V136I, I529L | weak biofilm | 1 |
| 405 | S83L,D87N,D678E | I663V | | S458A | weak biofilm | 1 |
| 361 | S83L,D87N | - | S80I, A84G, P401L | - | non biofilm producers | 5 |
| 361 | S83L,D87N | - | P401L | - | non biofilm producers | 3 |
| 361 | S83L,D87N | - | S80I, A84G | - | non biofilm producers | 2 |
| 361 | S83L,D87N | S464F | S80I, P461L | - | non biofilm producers | 1 |
| 361 | S83L,D87N | - | - | - | non biofilm producers | 1 |
| 167 | S83L,D87N | - | S80I, P577L | S458A | weak biofilm | 1 |
| 167 | S83L,D87N | - | S80I, P577L | S458A | Strong biofilm | 1 |
| 167 | S83L,D87N | - | S80I, D475R | S458A | weak biofilm | 1 |
| 167 | S83L,D87N | - | P577L | S458A | weak biofilm | 1 |
| 448 | S83L,D87N | – | S80I, L440R, | S458A | weak biofilm | 1 |
| 448 | S83L,D87N,R237H,D678E | – | S80I, L440R | S458T | weak biofilm | 1 |
| 448 | S83L,D87N,R237H,D678E | - | S80I, L440R | S458T | non biofilm producers | 1 |
| 448 | S83L,D87N | – | S80I, L440R, | S458A | non biofilm producers | 1 |
| 354 | S83L, D87N | E185D, S492N, A618T | S80I, E84G, R236L, D475E | V136I, V153I, I355T | Strong biofilm | 1 |
| 354 | S83L, D87N | E185D, S492N, A618T | S80I, E84G, R236L, D475E | V136I, V153I, I355T | moderate biofilm | 1 |
| 354 | S83L, D87N | A618T, T653A, I663V | S80I, R236L, D475D | V136I, V153I, I355T | moderate biofilm | 1 |
| 131 | S83L, D87N, A828S | – | S80I, E84V, I192V, A471G, D475R, Q481H | V136I, I529L | Strong biofilm | 1 |

**Table 3** (*continued*)

| ST | GyrA | GyrB | parC | parE | Biofilm formations | N |
|---|---|---|---|---|---|---|
| 131 | S83L, D87N, A828S | – | S80I, E84V, I192V, A471G, D475R, Q481H | V136I, I529L | moderate biofilm | 1 |
| 46 | S83L, D87N | – | S57T, S80I, D197E, K200N, D309E, D475E | Q428P | non biofilm producers | 1 |
| 46 | S83L, D87N | – | S57T, S80I, D157E, K159N, D309E, L343R, D475D | Q428P | weak biofilm | 1 |
| 10210 | S83L, D87N | – | S80I, D475R | S458A | non biofilm producers | 1 |
| 10210 | S83L, D87N | – | S80I | S458A | Strong biofilm | 1 |
| 10210 | S83L, D87N | – | – | – | non biofilm producers | 1 |
| 38 | D678E | – | D475E, Q695L | – | moderate biofilm | 1 |
| 38 | S83L, D87N, D678E | – | S80I, A90V, D475R | D463N | weak biofilm | 1 |
| 2011 | S83L, D87N, D678E | S492N, A618T, E655D | S80I, D475R | T172A, S458A | moderate biofilm | 1 |
| 457 | S83L,D87N,A863V | S492N | S80I | S458A | moderate biofilm | 1 |
| 1193 | S83L, D87N, D678E, A828S | – | S80I | L254Q | moderate biofilm | 1 |
| 1702 | S83L, D87N | – | S80I | S458A | non biofilm producers | 1 |
| 34 | S83L | – | – | – | non biofilm producers | 1 |
| 88 | S83L | D553E | – | A342T | non biofilm producers | 1 |

## Plasmidome in CREc urinary isolates

The most frequent plasmid replicon among the CREc isolates was IncFI (99.03%, 102/103), followed by Col (89.32%, 92/103) and IncI (19.42%, 20/103) (Table S5). Among the 103 urinary CREc isolates, we selected five representative strains, consisting of AMR0278, C032, C300, C359, and C439, for long-read sequencing to obtain complete genomes. Two different plasmid replicon types were identified in the 5 $bla_{NDM}$-harboring isolates (Fig. 3). The most frequent plasmid replicon type was IncFI (60%, 3/5), followed by IncFI/IncQ (40%, 2/5). The sizes of the three IncFI plasmids were in the range 86,537–107,837 bp, whereas the two IncFI/IncQ plasmids were in the range 108,269–134,243 bp.

Among the NDM-harboring plasmids, other antimicrobial-resistant genes ($bla_{OXA-1}$, $bla_{CTX-M-15}$, $bla_{TEM-1B}$, and *aac(6′)-Ib*) were detected in isolate nos. AMR0278, C300, C359, and C439 but not identified in C032 (Fig. 3A). Notably, the IncFI plasmid (no. AMR0278) carried three copies of the $bla_{NDM-5}$ gene in the same plasmid, and, to the best of our knowledge, this is the first such plasmid reported (Fig. 3). The plasmid carried *NDM-5* whose organization was close to the pM629-2-NDM5 of *E. coli* M629-2 isolated in Myanmar that also carried other antimicrobial-resistant genes ($bla_{OXA-1}$, $bla_{CTX-M-15}$, $bla_{TEM-1B}$, and *aac(6′)-Ib*) at a percentage identity of 99.97% (Fig. 3B). We identified several insertion sequences upstream of the three *NDM-5* genes in the IncFI plasmid (Fig. 3C).

This IncFI plasmid in the present study also carried $bla_{OXA-1}$, $bla_{CTX-M-15}$, $bla_{TEM-1B}$, and *aac(6′)-Ib*. IncFII plasmids carrying *NDM* genes have a restricted host range; however,

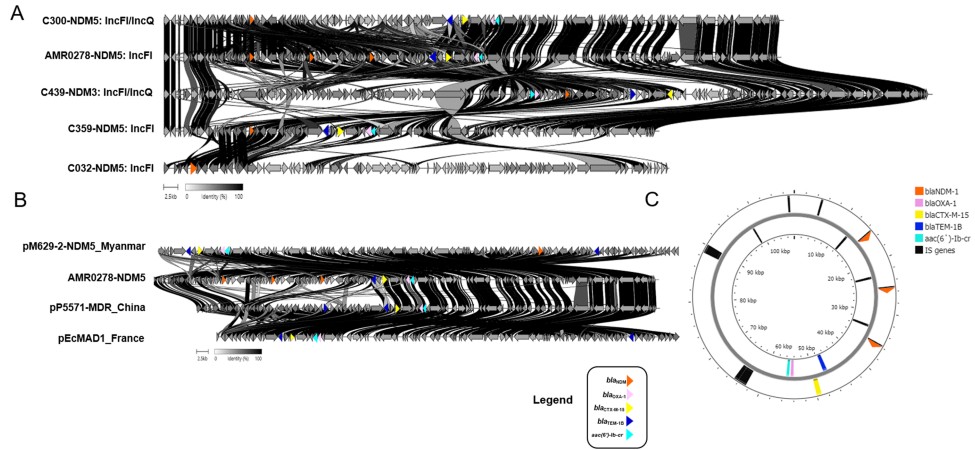

**Figure 3** **Protein level alignment of the $bla_{\text{NDM}}$ carrying plasmid in five CREc isolates.** Two different plasmid replicon types including IncFI and IncFI/IncQ were identified (A). Protein level alignment of the IncFI plasmid (no. AMR0278) carried NDM-5 with previously found in strains isolated in other country (B). Genes of interest are colored and defined in the legend. Alignments were made using clinker and clustermap.js and visualized using ApE v3.0.8. (C). Horizontal arrows indicate location, size, and direction of transcription, and visualized Proksee.

they are being increasingly reported (*Pitart et al., 2015*; *Fiett et al., 2014*). The IncFII-type plasmid (90 kb) reportedly co-carried $bla_{\text{NDM-5}}$ together with $bla_{\text{TEM-1}}$ and *rmtB* in a CREc isolate from a urine culture in Spain (*Pitart et al., 2015*). More recently, two $bla_{\text{NDM-5}}$-carrying IncF plasmids were present in ST167 *E. coli* strains isolated in China (*Feng et al., 2018*) and Italy (*Giufrè et al., 2018*).

Additionally, strain no. AMR0278 carried the $bla_{\text{OXA-181}}$ and *qnrS1* genes located on a nonconjugative ColKP3-type plasmid that was 51,478 bp long (Fig. 4). This plasmid contained type IV secretory genes upstream and the IS6 family genes upstream and downstream of the $bla_{\text{OXA-181}}$ and *qnrS1* genes. In addition, this plasmid showed high similarity to the plasmids of *E. coli* and *Klebsiella pneumoniae* from Switzerland, the Netherlands, Ghana, Egypt, France, China, and the United States (Fig. 4). The ColKP3 plasmid type harbored almost all classes of AMR genes, such as $bla_{\text{OXA-181}}$ and $bla_{\text{OXA-232}}$, and is common in *K. pneumoniae* (*Ragupathi et al., 2019*; *Weng et al., 2020*; *Naha et al., 2021*). The present study showed that a nonconjugative ColKP3-type plasmid carried the $bla_{\text{OXA-181}}$ and *qnrS* genes in an *E. coli* isolate.

A limitation of the present study was that we selected only five isolates to do complete genomes using the ONT platform, which may not have been representative of all the isolates in this study. Although we chose from the representative strains in each cluster of the phylogenetic tree, it could not cover all the studied strains that may have presented new characteristics. Therefore, further study with current CRE isolates should be undertaken and the results compared.

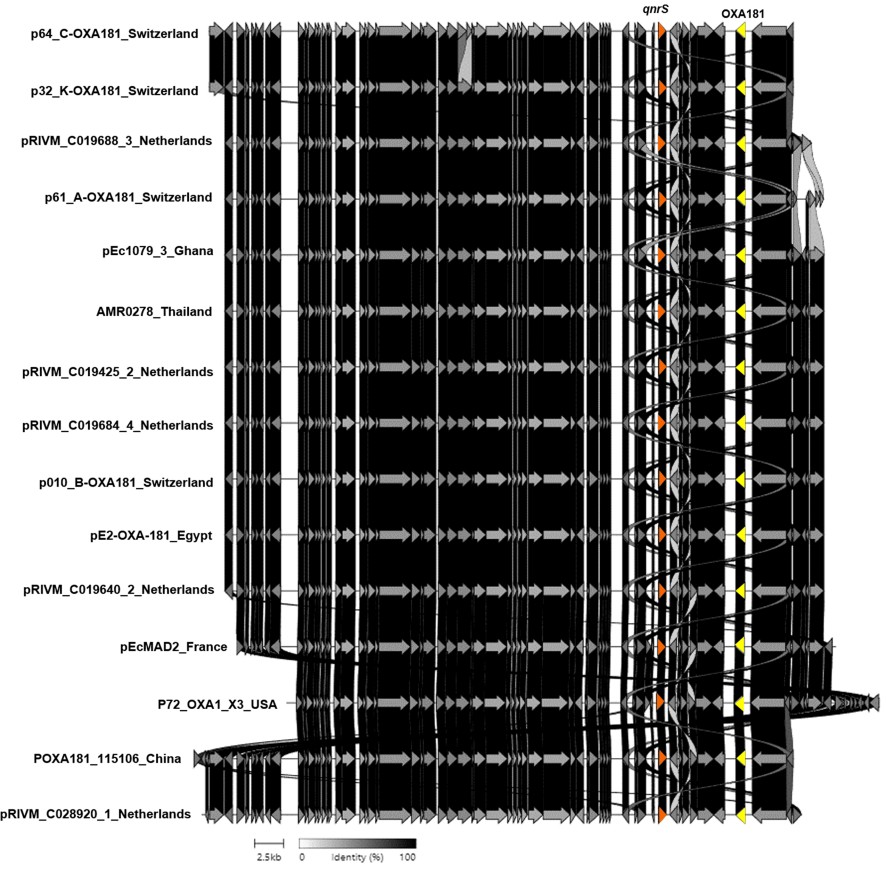

**Figure 4** Protein level alignment of the ColKP3-type plasmid carrying *bla*$_{\text{OXA-181}}$ and *qnrS* (no. AMR0278) with previously found plasmids in strains isolated in other country. Horizontal arrows indicate location, size, direction of transcription, and orientation of open reading frames. Alignments were made using clinker and clustermap.js.

## CONCLUSIONS

*NDM-5* and *CTX-M-15* were the predominant carbapenemase and ESBL genes, respectively, among the 103 urinary CREc isolates. The mutations in the QRDR of the *gyrA* and *parC* genes were the most predominant, followed by the presence of the PMQR determinant *aac(6′)-Ib*. The coexistence of *bla*$_{\text{CTX-M-15}}$ and *aac(6′)-Ib* was widely observed. Among the CREc isolates, resistance to ciprofloxacin and levofloxacin was greater than 90%. NFT maintained high sensitivity rates greater than 60%. They can be chosen as empirical antimicrobial treatments for uncomplicated UTIs. The results of this study highlighted the significant frequency of FQ resistance in CREc urinary isolates in Thailand, which is reflected in infection control and must be adopted to prevent further spread, emphasizing the need to expand antimicrobial drug resistance screening at hospitals. Therefore, to decrease the prevalence of FQ-resistant determinants in urinary CREc organisms in the future, continual epidemiologic surveillance and monitoring of antimicrobial prescriptions and consumption are required.

## ACKNOWLEDGEMENTS

We would like to thank Professor Constance Schultsz, Dr. Thomas Roodsant, Dr. Boas van der Putten, and Mr. Jaime Brizuela, Department of Medical Microbiology, University of Amsterdam, the Netherlands, for assistance in bioinformatic analysis and for providing critical suggestions. The authors would like to acknowledge Dr Andrew Warner of the Kasetsart University Research and Development Institute (KURDI), Bangkok, Thailand for assistance with English editing.

### Funding

This work was financially supported by the Office of the Ministry of Higher Education, Science, Research and Innovation; and the Thailand Science Research and Innovation through the Kasetsart University Reinventing University Program 2021. The funders had no role in study design, data collection and analysis, decision to publish, or preparation of the manuscript.

### Grant Disclosures

The following grant information was disclosed by the authors:
The Office of the Ministry of Higher Education, Science, Research and Innovation.
The Thailand Science Research and Innovation through the Kasetsart University Reinventing University Program 2021.

### Competing Interests

The authors declare there are no competing interests.

### Author Contributions

- Parichart Boueroy conceived and designed the experiments, performed the experiments, prepared figures and/or tables, authored or reviewed drafts of the article, and approved the final draft.
- Peechanika Chopjitt analyzed the data, authored or reviewed drafts of the article, and approved the final draft.
- Rujirat Hatrongjit analyzed the data, authored or reviewed drafts of the article, and approved the final draft.
- Masatomo Morita analyzed the data, authored or reviewed drafts of the article, and approved the final draft.
- Yo Sugawara analyzed the data, authored or reviewed drafts of the article, and approved the final draft.
- Yukihiro Akeda analyzed the data, authored or reviewed drafts of the article, contributed reagents, materials, analysis tools, and approved the final draft.
- Tetsuya Iida analyzed the data, authored or reviewed drafts of the article, and approved the final draft.

- Shigeyuki Hamada analyzed the data, authored or reviewed drafts of the article, and approved the final draft.
- Anusak Kerdsin conceived and designed the experiments, authored or reviewed drafts of the article, and approved the final draft.

## Human Ethics

The following information was supplied relating to ethical approvals (i.e., approving body and any reference numbers):

Ethical approval was obtained from the Ethics Committee of Osaka University Graduate School of Medicine, Osaka, Japan. The ethics approval number was 14468 and 22014. This study was conducted according to the principles of the Declaration of Helsinki. The need for informed consent was waived.

## Data Availability

The 15 urine isolates of CREc in this study from eight provinces in Thailand are available at GenBank: JAPIWS000000000, JAPIWR000000000, JAPDNK000000000, JAPIWN000000000, JAPIWQ000000000, JAPDNJ000000000, JAPIWP000000000, JAPIWO000000000, JAPDNI000000000, JAPDNH000000000, JAPDND000000000, JAPDNE000000000, JAPDNG000000000, JAPDNF000000000, and JAPDNL000000000.

## Supplemental Information

Supplemental information for this article can be found online at http://dx.doi.org/10.7717/peerj.16401#supplemental-information.

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
