# Peer review of "Fluoroquinolone resistance determinants in carbapenem-resistant Escherichia coli isolated from urine clinical samples in Thailand"

_PeerJ, doi:10.7717/peerj.16401_

## Round 0.1 · original submission · Major Revisions

Please revise the manuscript based on the comments from reviewers and improve the writing before resubmission.

Reviewer 1 ·

Basic reporting

This study assessed the fluoroquinolone resistance-determinants in 103 carbapenemresistant Escherichia coli isolated from urine clinical samples in Thailand. The authors had several significant findings and the manuscript is well-written

Experimental design

The study is well-designed and the method is described comprehensively. However, some statement such as the definition of CREc and the prevlaence of CREc in the orignial study needed to be added.

Validity of the findings

The associated results and discussion can be fairly supported by their findings.

Additional comments

1. Figure 1 is not needed and just mentioned in the text
2. Please add the definition of CREc.
3. Please briefly describe the original study and the prevelance of CREc among all urine isolates, and E. coli isolates

Reviewer 2 ·

Basic reporting

1> the article was poorly written in English and there were several grammatical errors and punctuation marks were missing in the text.
2> There were more than 55 references cited in the text, with the majority published before 2017. Please select informatic references with high quality.
3> The information in tables should be rearranged according to the description order, fimH description (line 181) in the first paragraph, but its data was displayed in table 3; For description about table 3, no text on fimH. If the strain serotypes were not important for FQ gene mutations, it could be represented with biofilm phenotype, MLST and Clermont phylogroup together, as the baseline characteristics.

Experimental design

1> why only 5 strains were sequenced by long-read sequencing technique, is there any unique feature for these five strains.

Validity of the findings

1> The description and tables in result section could be more summary, instead of raw representing the raw data. Especially for table 1, current form may be suitable for supplementary materials and another table with more conclusive data should be redraw.
2> Although there are a lot of genetic information analysis in urinary strains, it seemed that no relationship among them.
3> The discussion section should be rewritten based on the presentation order in result section, especially focused on the new findings.

Additional comments

The abbreviated forms were not displayed appropriately. Ssome abbreviated words had not been displayed as full form, such as E. coli, ExPEC. Repeat full form explanation for UTI, no need for WGS.

Reviewer 3 ·

Basic reporting

The present study used Whole genome sequencing to characterize fluoroquinolone resistance in CREc among urinary isolates in Thailand. The authors used 103 urine isolates over five years and performed a comprehensive analysis of the antimicrobial resistance profile of these isolates. The findings from the study emphasize the need to expand antimicrobial drug resistance screening at hospitals. The highlight of the study is the detailed characterization of the determinants of fluoroquinolone resistance and providing a detailed description of both PQMR and QRDR mutations in all isolates. The authors characterized the plasmid in these isolates and reported a plasmid with three copies of bla-NDM-5 gene. The high similarity of plasmids identified in this study with plasmids reported from other nations is of import and needs further discussion on the implications of these findings.
While the study addresses a major clinical issue of MDR organisms. discussion section would benefit from adding a sy=ubsection on how these findings can help develop/improve infection control policies and what changes do the authors recommend based on their findings.

Experimental design

The study is well designed with the use of various microbiological as well as bioinformatic tools to characterize drug resistance genes and mutation. The authors have also taken great care to ensure quality checks at each step of bioinformatic analysis and use reference genomes where necessary. However, providing the details of what Illumina platform was used in line 127 ( e.g., Nextseq1000 / Miseq / Nextseq 550) will be helpful.
Additionally, it would be helpful for the readers if the rationale/criteria were described in detail for selecting five isolates for long-read sequencing ( line 250).

Validity of the findings

The findings from the study are valuable to understand the antimicrobial drug resistance patterns and the study also underscores the growing threat of rapid evolution of MDR organisms and the need to identify these to aid treatments. The results also emphasize the need for efficient infection control strategies including genomic surveillance of the pathogens. however, the following changes may help to make the manuscript more appealing to the readers
Major comments.
- Figure 1, depicting sample collections from different provinces in Thailand, can be moved to supplementary data. Instead, it would be helpful for the readers if the data described in the result subheading “ genotyping profiles ..” is depicted graphically, say using a pie chart as Fig 1.
- In the subheading “Pathotying, virulence genes, and biofilm formation. ”, discussing the clinical significance of these findings, e.g., drawing conclusions based on correlations between biofilm formation capacity and the presence of AMR genes/mutations and its impact on treatment would be beneficial.

Additional comments

Overall, the manuscript would benefit from general English language editing to ensure the proper choice of words, spelling, meaning, and improved clarity.

Few examples below:

Line 70: omit ‘in’ from the sentence.
Line 79 – use of ‘however’ seems unnecessary.
Line 87: change to – ‘complicated cases’
Line 332: needs clarification.
Line 348-350: conclusion is unclear

·

Basic reporting

The authors explore the genomes of 103 carbapenem-resistant E. coli isolates acquired from a surveillance program and analyze the antimicrobial and virulence factors hosted by these isolates in the context of fluoroquinolone resistance. The isolates, some of which were a part of another published study, were sequenced via long and short-read sequencing, assembled, and analyzed using various bioinformatics tools for antimicrobial and virulence gene detection, pan-genome and phylogenetic analysis, etc. The authors position their contribution in the context of the use of fluoroquinolone (FQ) as the preferred antibiotic for the treatment of urinary tract infections (UTIs), increased occurrences of FQ-resistance in E. coli, and the scarcity of research studies characterizing the role of FQ for E. coli driven UTIs in Thailand.
The manuscript is quite weak in written English, particularly the Introduction section. The following sections of the review will elaborate on the concerns and other issues in detail.

Experimental design

Major Issues (in the order of their occurrences in the manuscript text):
1. The introduction section is all over the place. The authors should consider rewriting the entire section, and preferably get it reviewed by someone who is more proficient in written English. From a reader's standpoint, the paragraphs do not flow well, and becomes difficult to comprehend the relevance of sentences in the context of the topic and work to be followed.
2. Lines 59–61 are quite discordant from a reader's perspective. Since the emphasis of the work is on FQ resistance in E. coli for UTI treatment, it would be better suited to introduce E. coli as one of the primary causal factors for UTI, and then expand it a little further about the treatment strategies used for E. coli driven UTIs and maybe the healthcare burdens associated to it.
3. Line 63: why is FQ not mentioned earlier in the list of drugs used for UTI treatment?
4. Line 64: “… with unnoticeable side effects, …” This is not true! Studies have reported instances of rare but serious side effects and have been acknowledged by the USFDA as well:
- doi.org/10.1016/j.cmi.2019.10.016
- doi.org/10.1177/2324709614545225
5. Several instances of incorrect grammar in sentence construction, incorrect spelling of ‘Enterobacterales’, incomplete sentences (deleted words) merged into another sentence (line 167), and sentences containing only the adverbial or relative clauses (i.e., incomplete sentence constructions).
6. The literature review for FQ resistance (lines 72–78) refers to studies from 2009. In the context of bioinformatics, these are older studies (>10 years ago). It is recommended that the authors cite relatively recent works more. In a way, that also highlights the contemporary relevance of this work (given the existence of several other works published in multiple esteemed venues, the authors position their work to be important and relevant for the field as a whole).
7. Line 79: this claim seems to be the central emphasis on the shortcomings of the existing literature and the relevance of the proposed work. However, this claim comes out of nowhere (without introducing the shortcomings/limitations of existing studies, the need to fill this gap in the literature, etc.). The authors should rephrase this line/paragraph to give some context as to why is this important.
8. Line 83–84: There are several studies published in the last 7–8 years on CRE and CoRE. It doesn't make a lot of sense to come across lines 83-84: “Two recent studies provided information on the molecular epidemiology and characteristics of CRE and CoRE”. In case these two studies were particularly helpful in the context of this work, those aspects should be expanded and described here appropriately. Just mentioning two studies provided the information when there are several others on similar topics doesn't make a lot of sense.
9. Line 137–138: The authors have used Unicycler for hybrid genome assembly. Given that the authors have access to both long and short-read sequences, I request the authors also try out other hybrid assembly approaches and choose the one that provides the best genome assembly quality and completeness. This manuscript enlists some of the current hybrid assembly approaches which are applicable to the authors’ setup as well: doi.org/10.1186/s12864-021-07702-2
10. Line 144: it is recommended to check against CARD as well given that it is being increasingly accepted by the community as the standard and well-maintained database of ARGs and virulence genes: doi.org/10.1093/nar/gkac920
11. Phylogenetic and pan-genome analyses: It is recommended to have a deeper dive into the pan-genome and phylogenetic analysis. Please refer to the following works and use some of the relevant approaches in this work:
- doi.org/10.1093/bioinformatics/btac844
- doi.org/10.1093/bioinformatics/btv421
- doi.org/10.3389/fmicb.2021.714284
- doi.org/10.1186/s13059-020-02042-y
- doi.org/10.7717/peerj-cs.20

Validity of the findings

Major Issues (in the order of their occurrences):
1. Line 184: The authors need to first provide the elementary statistics about the genotype-phenotype counts, and the assembly quality stats (mean length, N50, etc.)
2. Line 187–188: The title of the manuscript mentions FQ-resistance determinants in CREc, which implies that the proposed work intends to highlight the genotypic factors associated with or causal for the phenotypic resistance. However, lines 187–188 mention that 100/103 isolates were FQ-resistant. With such dichotomy in the phenotypic groupings, it is statistically improbable to identify the actual factors statistically significantly associated with FQ resistance. To present the manuscript with such phenotypic class imbalance, the authors should reconsider the central emphasis of their work and adjust their manuscript title and other relevant sections accordingly.
3. Line 188: “CREc isolates grouped into…”: How was this grouping made?

Reviewer 5 ·

Basic reporting

English language throughout the text needs improvement. For example in lines 213 214 the authors state:
ESBLs screening test showed that 77.67% (80/103) were ESBL producers, with 97.50% (78/80) were FQs resistance
Instead of:
ESBL screening test showed that 77.67% of the isolates were ESBL producers and among them 97.50% were also resistant to fluorquionolones.

In lines 220 221 the authors state:
The carbapenemase genes were identified in NDM-5 (74.76%, 77/103), followed by NDM-1 (17.48%, 18/103) (Table 1).
Instead of:
The carbapenemase genes identified were blaNDM-5 (74.76%, 77/103), followed by blaNDM-1(17.48%, 18/103) (Table 1).

Experimental design

The manuscript focuses on fluoroquinolone resistance determinants of 103 carbapenem resistant E. coli isolates The majority of them (88) have been previously described elsewhere (Takeuchi D, Kerdsin A, Akeda Y, Sugawara Y, Sakamoto N, Matsumoto Y, Motooka D, Ishihara T, Nishi I, Laolerd W, Santanirand P, Yamamoto N, Tomono K, Hamada S. Nationwide surveillance in Thailand revealed genotype-dependent dissemination of carbapenem-resistant Enterobacterales. Microb Genom. 2022 Apr;8(4):000797. doi: 10.1099/mgen.0.000797. PMID: 35438076; PMCID: PMC9453063.) Their genetic properties and phylogenetic analysis have been already discussed in the abovementioned study.
The authors report 15 new sequences that have not been published before. They analyze the sequences collectively but the design is not clear. The authors have also used long read sequencing for 5 isolates, still not clear why this approach was used, there is no discussion on the differences of short read and long read sequencing.

Validity of the findings

No comment.

Additional comments

Ethical approval was not in English, I could not read it.

---

## Round 0.2 · Major Revisions

Please address the comments from both editors and revise the manuscript accordingly.

**Language Note:** The review process has identified that the English language must be improved. PeerJ can provide language editing services - please contact us at copyediting@peerj.com for pricing (be sure to provide your manuscript number and title). Alternatively, you should make your own arrangements to improve the language quality and provide details in your response letter. – PeerJ Staff

Reviewer 2 ·

Basic reporting

1. “Almost all ExPEC isolates in this study were mainly belonging to the phylogenetic groups B2 and D and carried various virulence factors when compared to the other phylogenetic groups.” It seemed that the authors wanted to say more virulence factors in phylogenetic groups B2 and D, but this information could not be obtained from the tables or figures.
2. “Although most urinary CREc in this study could not be classified, they seem to be commensal or non-pathogenic pathotypes.” Did the author mean that most strains in this study could not be classified into any class?, since the information could not be read from table 2 directly.
3. “In the biofilm assay, 50.49% (52/103) of the urinary CREc isolates showed weak-to-strong biofilm formation (Table 2 and S4).” In table 2, there is no information about biofilm formation.

Experimental design

1. There is no information about definition for APEC, positive if one or ten genes present?
2. What method was used to define the resistance to carbapenem antibiotics or the presence of carbapenemase.
3. The authors gave the results of susceptibility against FQs and NFT, but there is no information how the assay is performed in the method section.

Validity of the findings

1. Table notes for table 1 was not consistent to the information in Table 1. No data on “I” in Table 1, but mentioned in notes. Also the “PMQR”, “QRDR”,“NFT”
3. “Among the four investigated virulence factor classes,”, there were only three classes mentioned in the result and method section.
4. “NDM-5 and CTX-M-15 are the predominant carbapenemase and ESBL genes, respectively, among the 103 urinary CREc isolates.” NDM-5 and CTX-M-15 seemed to mean the genes’ names, which should be italic in scientific format.
5. In the discussion section, no limitation was mentioned. Although the author mentioned five isolates for WGS were representative strains based on phylogenetic tree, I am afraid that the representativeness was doubtful. The author could state this in limitation section.
6. Inconsistent number of enrolled strains between supplementary figure 1 legend (n=104) and the main text (103).

Reviewer 5 ·

Basic reporting

English still needs improvement. The authors state that professional English language services have edited the manuscript, but it doesn't seem the case.

Experimental design

This is an additional study that needs to take into consideration the work done in the previous one. As such all authors of the first study need to be included either as co-authors or collaborators. Otherwise written permission that their collective work can be used in this study must be included.

Validity of the findings

No comment.

Additional comments

The authors could not provide ethical consent in English. I suggest they translate the Japanese version of the consent into English and provide the translated copy additionally.

---

## Round 0.3 · accepted · Accept

I confirm that the authors have addressed all of the reviewers' comments, and the current version is ready for publication.

Reviewer 2 ·

Basic reporting

no comment

Experimental design

no comment

Validity of the findings

no comment

Additional comments

no comment

Reviewer 5 ·

Basic reporting

Professional English has been used throughout the text, and the authors have improved the manuscript as suggested in the review process.

Experimental design

The authors have provided a reasonable explanation of the selection of strains from the previous study. They have also provided written permission from the corresponding author of the previous study, on behalf of all the authors.

Validity of the findings

No comment

Additional comments

The authors have provided translated copies from Japanese to English language, of the ethical consents, as requested during the review process.